Journal of Data-centric Machine Learning Research (2024)        Submitted 03/24; Revised 08/24; Published 10/24

# Properties of Alternative Data for Fairer Credit Risk Predictions

**Jung Youn Lee**[*]                                             JUNGYOUN.LEE@RICE.EDU
*Jones Graduate School of Business*
*Rice University*
*Houston, TX 77005, USA*

**Joonhyuk Yang**[*]                                             JOONHYUK.YANG@ND.EDU
*Mendoza College of Business*
*University of Notre Dame*
*Notre Dame, IN 46556, USA*

**Reviewed on OpenReview:** *https://openreview.net/forum?id=euJwgThZom*

**Editor:** Yang Liu

## Abstract

In the consumer lending market, women tend to have lower access to credit than men, despite evidence suggesting that women are better at repaying their debts. This study explores the potential impact of leveraging "alternative data," which traditionally has not been used by financial institutions, on credit risk predictions between men and women. By leveraging unique data on individuals' credit card default behaviors and their purchase behaviors at a supermarket, we simulate a credit card issuer's credit scoring process. In the absence of supermarket data, the algorithm's predictive accuracy for women is about 2.3% lower than that for men. We then integrate data from each of the 410 product markets within the supermarket into the algorithm and measure the changes in the gender gap in predictive accuracy. We find a wide variation in both direction and magnitude in the incremental gender gap, ranging from -142% to 70% compared to the baseline. These findings highlight that leveraging alternative data from a non-financial domain can lead to fairer credit outcomes, but only under certain conditions. We characterize the conditions by identifying two data properties: the capacity to proxy gender and the relative amount of creditworthiness signals data provide for each gender.

**Keywords:**  alternative data, proxy discrimination, gender gap, credit scoring, algorithmic fairness.

## 1 Introduction

In many parts of the world, women tend to have lower access to credit compared to men (World Bank, 2022). Even when women secure credit, they face higher interest rates or lower loan amounts (Cheng et al., 2011; Van Rensselaer et al., 2014; Blascak and Tranfaglia, 2021). This pattern persists despite evidence suggesting that women are better at repaying their debts than men (Goodman et al., 2016; Perrin and Weill, 2022). One potential explanation

---

*. Equal contribution.

for this gender gap in credit access is the gender data gap. Women are less likely to interact with formal financial institutions (Martin and Longa, 2011; Schmitz, 2014), resulting in "thinner" credit histories compared to men. Consequently, traditional credit scores, which are constructed primarily based on credit histories, often either do not exist for women or provide less precise and reliable signals of creditworthiness for them. This relative deficiency in data for women impairs lenders' ability to accurately evaluate the creditworthiness of female borrowers and extend credit to creditworthy women and on fair terms.

Credit scoring that leverages "alternative data" presents an opportunity to close the gender data gap, thereby closing the gender gap in credit access. Alternative data refers to non-traditional data sources that provide insights into the creditworthiness of a wider spectrum of individuals, even those outside the formal financial system. This has the potential to ensure more equal representation of diverse groups and reduce disparities in the evaluation of creditworthiness. The value of various alternative data sources is increasingly recognized in both the public and private sectors, spanning across monthly payment records for rent, utilities, and phone bills (Experian, 2021; Fannie Mae, 2021), cash flow data (FinRegLab, 2019), digital footprints (Berg et al., 2020), mobile phone usage data (Björkegren and Grissen, 2020), education history (Odinet, 2018), text data (Netzer et al., 2019), and retail purchase data (Lee et al., 2023).

However, the impact of utilizing alternative data on the gender gap remains uncertain. In many countries and jurisdictions, fair lending regulations prohibit lenders from considering gender at any stage of the credit evaluation process. Nonetheless, even when data lack explicit references to gender, certain variables or combinations of variables within the data may exhibit correlations with gender, effectively serving as a proxy for group membership (Dwork et al., 2012; Yang and Dobbie, 2020; Ascarza and Israeli, 2022). This phenomenon, known as indirect or proxy discrimination (Prince and Schwarcz, 2019), can unintentionally result in gender-based disparities. Similarly, in the context of racial disparities, Odinet (2018) highlights that the use of education-based alternative data, such as transcripts, standardized test scores, and college majors, by fintech lenders in student lending can "lead to a greater class divide," as these data points may strongly correlate with race, ethnicity, and household income. This concern has been exacerbated by the use of machine learning algorithms capable of uncovering hidden relationships among variables (Fuster et al., 2022). In response, some propose statistical solutions to purge predictions of both direct and proxy effects of protected variables, although their implementation necessitates access to and utilization of protected variables (Yang and Dobbie, 2020).

In this paper, we explore the potential disparate impact of leveraging alternative data for credit scoring on men and women, with a focus on retail purchase data as a form of alternative data. Our analysis benefits from a unique data set consisting of first-party data from two firms owned by a conglomerate: a credit card issuer and a large-scale supermarket chain.[1] Using a customer identifier, we can merge data from both firms at the individual level, which allows us to build a hypothetical credit scoring algorithm that leverages supermarket transaction data.[2] Another valuable aspect of our data set is that the conglomerate

---

1. As our data is protected by a non-disclosure agreement with the data provider, we are unable to publicize the data. In Appendix A, we provide summary statistics for the features derived from the data.
2. In the discussion section, we explore the feasibility of employing alternative data for credit scoring, considering the implementation methods that can mitigate privacy and fairness concerns.

collects gender information from its members, although this information is not used for the issuer's credit decisions. This feature enables us to empirically explore the impact of using alternative data separately for men and women. This is a significant advantage, as empirical research on gender-based disparities in consumer credit markets has been limited. This limitation is largely due to the fact that gender is a protected class in many countries, including the United States, the United Kingdom, Canada, the European Union, Australia, and India. Consequently, lenders are often prohibited from collecting applicants' gender information or linking it with credit data.

In the remainder of this paper, we leverage this unique data and investigate the association between the properties of alternative data and the resulting gender disparities in credit risk prediction accuracy. We begin by providing a conceptual framework and details on our data and empirical strategy, followed by results and discussion.

## 2 Conceptual Framework

Our key metric for assessing the gender gap in credit access is the disparity in *predictive accuracy* of credit scoring algorithms between men and women, which is widely used to measure algorithmic bias and fairness (Feldman et al., 2015; Sculley et al., 2018). In our context, predictive accuracy indicates the algorithm's ability to distinguish between those who would default and those who would not. A lower predictive accuracy implies a higher rate of approving risky applicants (false approvals) and/or a higher rate of rejecting creditworthy applicants (false denials). Both false denials and false approvals can amplify gender disparities in credit access and undermine consumer welfare in the long run. For example, rejecting creditworthy women could preclude them from accumulating wealth and making long-term investments in housing, career development, and education, which would hinder their future credit access. On the other hand, extending credit to risky women may lead to higher default rates among women, which in turn, can make it even more challenging for the average woman to secure credit. In essence, our metric quantifies the difference in the algorithm's error rates or the extent of "unfair" predictions that cannot be rationalized by inherent gender differences in creditworthiness (Dwork et al., 2012).

To fix ideas, consider a scenario in which a lender is making loan approval decisions for credit applicants. The lender's decisions rely on a credit scoring algorithm, $f(Y|X)$, designed to predict the credit risk $Y$ of applicants. In the baseline scenario, the algorithm predicts applicants' credit risk only based on traditional financial data $X$ (e.g., income, occupation, credit scores). Note that the algorithm does not incorporate gender as input, as the lender does not have access to this information and/or it is legally prohibited. Due to fair lending laws, the explicit use of protected variables such as $G$ in credit evaluation is often prohibited. The unavailability of gender information prevents lenders from giving gender-specific weights within the algorithm or constructing gender-specific algorithms. However, we as researchers observe the applicants' gender $G \in \{m, w\}$, where $m$ represents men, and $w$ represents women. The baseline gender gap can be written as $\text{Gap}(X) \equiv u[f(Y|X); G = m] - u[f(Y|X); G = w]$, where $u(\cdot; G)$ is a function that measures the algorithm's predictive accuracy for the respective gender group.

The quantity of our interest is the *incremental* gender gap, denoted by $\Delta^A$Gap, introduced by adding alternative data $A$ to the lender's information set. $\Delta^A$Gap can be written as follows:

$$
\begin{aligned}
\Delta^A\text{Gap} &\equiv \text{Gap}(X, A) - \text{Gap}(X) \\
&= \{u[f(Y|X, A); G = m] - u[f(Y|X, A); G = w]\} - \{u[f(Y|X); G = m] - u[f(Y|X); G = w]\} \\
&= \{u[f(Y|X, A); G = m] - u[f(Y|X); G = m]\} - \{u[f(Y|X, A); G = w] - u[f(Y|X); G = w]\} \\
&= \Delta^A u[f(Y); G = m] - \Delta^A u[f(Y); G = w].
\end{aligned}
$$
(1)

The last line of Equation 1 indicates that the size and direction of the incremental gender gap depend on two terms, $\Delta^A u[f(Y); G = m]$ and $\Delta^A u[f(Y); G = w]$, which represent the incremental predictive accuracy of the algorithm introduced by $A$ for men and women, respectively.

We hypothesize that $\Delta^A$Gap will depend on the interplay between two characteristics of alternative data $A$. The first attribute is "signal disparity," which denotes the difference in creditworthiness signals present in $A$ for men and women. For example, consider a scenario where high supermarket expenditure on cigarettes is a strong signal of future default among women but holds little predictive value for men because nearly all men in the sample buy cigarettes. In this case, integrating cigarette expenditure data into credit scoring would disproportionately improve predictive accuracy for women, leading to a negative incremental gender gap.

The second characteristic of $A$ that can impact $\Delta^A$Gap is its capacity to proxy $G$, which we refer to as "class separability." As mentioned earlier, the lender does not have access to $G$. Therefore, credit scoring algorithms are trained on pooled data across genders, potentially overlooking gender-specific differences. Instead, they learn the *average* relationship between individuals' behaviors and credit risk, treating both men and women as a combined group, which may dilute the potential for differential improvement in predictive accuracy.[3] Continuing with the previous example, the algorithm may over-predict the credit risk of men who purchase cigarettes while under-predicting the risk of women who do the same.

However, even in the absence of variable $G$ as an input, the algorithm may still infer the presence of distinct gender groups from $A$ if the data exhibits correlations with $G$ and thus serves as a proxy for them. For example, suppose that all women in our sample make purchases from the female hygiene products category, and higher category expenditure is associated with lower credit risk for women. Suppose none of the men make purchases from this category. In such a case, the algorithm may interpret the data as follows: given positive category expenditure (i.e., the individual is a woman), there exists a correlation between credit risk and category expenditure. Consequently, utilizing expenditure data from the female hygiene products category might improve predictive accuracy only for women. In other words, when $A$ can construct a strong proxy for $G$, even a gender-blind algorithm becomes capable of identifying gender-specific relationships between supermarket expenditure and creditworthiness.

Below, we empirically investigate how these two data properties are associated with the resulting incremental gender disparities.

---

3. All credit scoring algorithms presented in this paper are trained on data pooled across genders.

## 3 Method

### 3.1 Data

Our empirical analysis utilizes unique data sets provided by an anonymous conglomerate operating in several countries in Asia and Africa. This conglomerate consists of multiple business units, including a credit card issuer and a large-scale supermarket chain. The credit card issuer provides general-purpose credit cards accepted at any merchants with associated processing networks (e.g., Mastercard, Visa). The supermarket chain sells a range of products across various categories, including groceries, household supplies, clothing, and other general merchandise.

We leverage first-party data from both the credit card issuer and the supermarket chain. From the credit card issuer, we obtain three types of account-level data for approved card-holders: socioeconomic variables, credit scores, and credit card repayment history. As part of the credit card approval process, the issuer collects self-reported information from the credit card applicants, including the number of dependents, monthly income, employment status, and occupation. This information is supplemented with credit scores purchased from the local credit bureau. In addition, we have access to the cardholders' repayment history between June 14, 2017 and June 13, 2019. This data records the presence and duration of any outstanding debt owed by the cardholder, as well as monthly payment status (e.g., normal payment, over-payment, no payment).

From the supermarket chain, we obtain the scanner panel data on transactions between January 1, 2017, and December 31, 2018. This data records purchases made with the supermarket's loyalty cards across different stores within the chain, regardless of the payment method. The data includes several variables such as timestamp, item code, item hierarchy information, brand code, quantity purchased, price listed, price paid, customer identifier, transaction identifier, and store identifier.

Using the hierarchical structure of the supermarket's product offerings, we define "markets" within the supermarket data, where a market represents a specific segment of products sold. Specifically, the hierarchical categorization comprises 42 sections (e.g., beverages),[4] which can then be broken down into a total of 410 product categories (e.g., mineral water). Each product category can be further divided into a total of 2,184 product subcategories (e.g., flavored mineral water).[5]

A key feature of our empirical strategy is to treat each of the 410 product categories as a distinct, independent market, each serving as a unique source of alternative data. This allows us to evaluate 410 distinct scenarios, where we observe variations in both data properties and the resulting incremental gender gap. By doing so, we aim to explore any systematic relationships between the two variations. This approach is in contrast with prior studies that typically focus on a single source of alternative data. It is important to acknowledge that our approach may not perfectly align with how financial institutions

---

4. The full supermarket transaction data set contains data from 52 sections. We exclude 10 sections that cover transactions at concession stands and third-party vendors, such as consignment stores, flower shops, and music stores. Transactions from these excluded sections account for about 0.085% of the total number of transactions and 0.712% of the total revenue during the sample period.

5. Within our observations period, the sample consumers made at least one transaction in each of these 410 product categories. For details on these markets, refer to Appendix A.

would utilize such data. In practice, institutions might seek to enhance overall predictive accuracy by leveraging the entirety of the multi-market data concurrently. However, our goal is not to provide findings specific to supermarket data. Instead, we aim to consider each product market as an individual data source characterized by unique treatments (i.e., data properties) and outcomes (i.e., incremental gender gap).

We use a customer identifier to merge the credit card repayment history and the supermarket loyalty card data at the individual level.[6] Prior to this merge, we filtered the data set to include only individuals who meet the following four criteria: (1) they have self-reported gender information, which is sourced from the conglomerate that collects this data through its company-wide membership program, as opposed to being sourced from either the credit card issuer or the supermarket chain, (2) they possess a single credit card with the credit card issuer, (3) they have one supermarket loyalty membership, and (4) they made at least one purchase at the supermarket during the sample period.

After applying these criteria, our final sample consists of 30,005 individuals, among whom 25,827 (86%) are men and 4,178 (14%) are women. Men and women differ significantly across various dimensions. Specifically, compared to the median man, the median woman has a lower income ($6,673 USD for women vs. $7,850 USD for men; $p=.005$) and a slightly lower credit score (720 for women vs. 728 for men; $p=.019$). Importantly, women are more likely to default than men: 8.0% of men and 9.7% of women defaulted during the sample period. Detailed summary statistics about the final sample are available in Appendix A.

### 3.2 Credit Scoring Simulations

To explore the potential impact of alternative data sources on the gender gap, we consider a hypothetical scenario in which the issuer develops a credit scoring algorithm to predict the credit card repayment behaviors (or default) for its applicants. This algorithm is trained using the repayment history and characteristics of *past* applicants who were approved and enrolled in the issuer's credit cards (i.e., existing cardholders). Such historical data exclude information about individuals who either did not apply, applied but were rejected, or were approved but chose not to enroll. While the use of historical data for algorithm construction mirrors a prevalent industry practice, one that is also supported by our data provider, we acknowledge that this approach may not fully uncover the potential of utilizing alternative data for a broader population of consumers within the consumer credit market. Instead, our focus is more narrowly directed toward individuals who were likely to be granted credit by the issuer's existing credit scoring algorithms.

We first split the sample period into two distinct, non-overlapping periods: Periods 1 (15 months) and 2 (15 months). We then adopt the perspective of the issuer who scores applicants at the beginning of Period 2 based on consumer data available from Period 1. Importantly, we simulate the issuer's scoring process under different information sets by

---

6. One potential concern is that household members might share supermarket loyalty cards, leading to a mismatch between the shopper's gender and the cardholder's gender. However, we are not overly concerned about this issue because our main analysis focuses on across-product category comparisons while holding other factors constant, including the sharing of supermarket loyalty cards. This approach allows us to attribute the variation in the incremental gender gap across product categories solely to differences in shopping behaviors across categories.

constructing two different sets of input features for the credit scoring algorithm. First, to establish a baseline gender gap in the absence of alternative data sources, we train a scoring algorithm only using traditional variables available to us. These variables include traditional credit scores purchased from the bureau, income, employment status and occupation, and number of dependents. Second, to assess the impact of incorporating alternative data sources into credit scoring, we train a series of credit scoring algorithms, each of which incorporates data from one of the 410 product categories sold in the supermarket or "markets," in addition to the aforementioned traditional data sources.

We transform the supermarket data into usable inputs for the algorithms by creating a set of variables for each market that quantifies an individual's spending within the product subcategories belonging to that market (product category). For instance, when considering the carbonated drinks category, which includes seven subcategories, such as cola and non-alcoholic beer, we create seven features including cola expenditure and non-alcoholic beer expenditure. We also explore alternative approaches to feature engineering, such as total category expenditure and binary indicators for subcategory purchases, and find qualitatively similar results. Details on the resulting features and results based on alternative feature engineering are available in Appendix A.

To train the algorithms, we randomly split our final data set consisting of 30,005 consumers into a training set (70%) and a test set (30%). In doing so, we employ a stratified sampling approach to maintain an identical share of defaulters in both sets. Then, we train a binary classifier on the training data using XGBoost (eXtreme Gradient Boosting; Chen and Guestrin, 2016) to predict whether an individual will default, where its hyperparameters are tuned through a Bayesian optimization method with 10-fold cross-validation (Wilson, 2021). We also re-scale the gradient for the positive class (i.e., default) and over-correct errors related to the class in order to mitigate the potential impact of class imbalance (i.e., having significantly more non-defaulters in the data than defaulters) on the predictive performance of our algorithms.

The algorithm's out-of-sample predictive performance is evaluated on the test set. This entire process is iterated 1,500 times, each with different random training/test data splits and hyperparameter tuning. The averaged prediction outcomes from these iterations are reported. Our findings remain robust to alternative modeling choices, including a larger number of iterations and the use of alternative binary classifiers like logistic regression and elastic-net regularized generalized linear models. Further details on the algorithm training and validation procedures can be found in Appendix A.

### 3.3 Measure of Predictive Gender Gap

We define the gender gap as the disparity in predictive accuracy between men and women in the credit scoring algorithm. It is important to note that our metric does not directly measure gender disparities in actual credit *decisions*; rather, it measures disparities in the quality of *predictions* produced by the algorithm. Exploring the impact of alternative data on credit decisions necessitates considering a specific decision rule or the issuer's objective (e.g., threshold score for credit approvals) (Barocas et al., 2017). Additionally, issuers may integrate outputs from multiple scoring algorithms into their final decisions, which are not available to us. Instead of making assumptions about such a decision process, we focus on

the gender disparities in the algorithm's predictive accuracy, which can hinder the efficient distribution of credit between genders, regardless of the specific decision-making framework in place. This approach allows for a broader evaluation of the impact of alternative data.

In line with this objective, we use the AUC (Area Under the Curve) metric to measure predictive performance (e.g., Dressel and Farid, 2018; Gardner et al., 2019). The AUC metric characterizes a trade-off between the algorithm's true positive and false positive rates across *all* possible classification thresholds. The AUC metric is closely connected to one of the notions of fairness in machine learning known as "equalized odds," which requires that individuals in all groups have equal true positive rates and equal false positive rates (Hardt et al., 2016). Another advantage of using the AUC metric is its robustness to imbalanced data (Jeni et al., 2013). This feature is beneficial in our setting where 8.2% of the sample consumers are defaulters and 91.8% are non-defaulters.

The value of AUC typically ranges between 0.5 and 1, where an AUC value of 1 corresponds to a perfect prediction, while an AUC value of 0.5 results from a random guess of a binary outcome variable. To quantify the gender gap, we first train an algorithm on pooled data and take the algorithm to make out-of-sample default predictions separately for men and women. The gender gap is then defined as the difference between the gender-specific AUCs, i.e., $\text{AUC}_m - \text{AUC}_w$.

We emphasize that our metric is designed to capture disparities in *predictive* performance that cannot be explained by potential disparities in true creditworthiness between men and women. Within our specific context, women might experience a higher rate of credit denials than men as they are riskier (default rate is 9.7% for women compared to 8.0% for men). However, this disparity may not necessarily be influenced by the differential predictive accuracy of the algorithm. For instance, fair lending laws in the United States, such as the Equal Credit Opportunity Act and the Fair Housing Act, do not categorically deem it unlawful for policies or practices to disproportionately affect individuals in protected classes; this may be considered acceptable under these laws if there exists a valid "business necessity" justification. By contrast, our metric focuses on capturing the algorithm's precision in classifying observations as defaulters or non-defaulters, conditional on actual default outcomes.

## 4 Results

### 4.1 Baseline Gender Gap

When relying solely on traditional data sources, the credit scoring algorithm generates less accurate predictions for women. The out-of-sample AUCs are 0.675 for men and 0.660 for women, leading to a baseline gender gap of 0.0156. In other words, the baseline AUC for women is about 2.3% lower than that for men.[7] Our baseline gender gap is comparable, at least in terms of magnitude, to findings reported in the literature. For example, Björkegren and Grissen (2018) report that when using only credit bureau information to predict default

---

7. Based on 1,500 repeated random samplings, a two-sided, paired Student's t-test indicates that the difference is statistically significant at the 5% level ($p < .001$). However, we note that in our case, this type of statistical precision measure can be misleading due to the dependency across samples (Dietterich, 1998; Vanwinckelen and Blockeel, 2012).

in a middle-income South American country, the out-of-sample AUC is about 0.580 for men and 0.600 for women, resulting in a gender gap of -0.020.

The gender gap could be at least partially influenced by an imbalance in our training data: women represent 14% of the final sample. This data imbalance could adversely affect the algorithm's accuracy for the minority group. However, assessing this training data imbalance itself presents challenges in practice, as regulations prohibit lenders from collecting or employing applicants' gender (or any other protected attributes) during the credit assessment process. Although there could be various explanations for the gender-imbalanced data, including historical societal biases against women, our goal is not to uncover the root causes of the gender gap generated by traditional data sources. Instead, we take the baseline gap as given and focus on understanding the *incremental* gender gap introduced by the use of alternative data sources, which we explore in the following section.

## 4.2 Incremental Gender Gap from Alternative Data Sources

Figure 1 presents the distribution of gender gaps resulting from the incorporation of alternative data sources. The blue dashed line represents the baseline gender gap of 0.0156, while the gray solid line indicates the absence of the gender gap (i.e., equal out-of-sample AUC for men and women). The dots positioned above the blue dashed line indicate that relative to the baseline, the use of the corresponding data source serves to *widen* the gender gap, disadvantaging women to a greater extent. On the other hand, the dots placed below the blue dashed line but above the gray solid line indicate that the inclusion of the respective data source *reduces* the gender gap compared to the baseline. The dots positioned below the gray solid line indicate that using the corresponding data source reverses the direction of the gender gap: the predictive accuracy for men becomes lower than that for women, resulting in a gap favoring women.

The results reveal a substantial variation both in the direction and magnitude of the incremental gender gap across data sources relative to the baseline gap. Out of the 410 data sources explored, approximately 25% contribute to widening the existing gender gap, further disadvantaging women. About 50% lead to a reduction in the gap. 25% of the data sources do not significantly impact the gap, and only a few (3 out of 410) lead to a reversal of the direction of the gap.[8]

## 4.3 Determinants of the Incremental Gender Gap in Predictive Accuracy

To explain why the use of certain data sources can either widen or narrow the predictive gender gap, we explore two distinct properties of these data sources.

### 4.3.1 PROPERTY 1 (SIGNAL DISPARITY): DIFFERENTIAL AMOUNTS OF CREDITWORTHINESS SIGNALS

We posit that a larger incremental gender gap is likely to emerge when a data source exhibits a more disproportionate ability to predict creditworthiness for one gender than the other,

---

8. Based on a series of two-sided, paired Student's t-tests, the incremental gender gaps are statistically significant at the 5% level for 286 out of the 410 sources (roughly 70%).

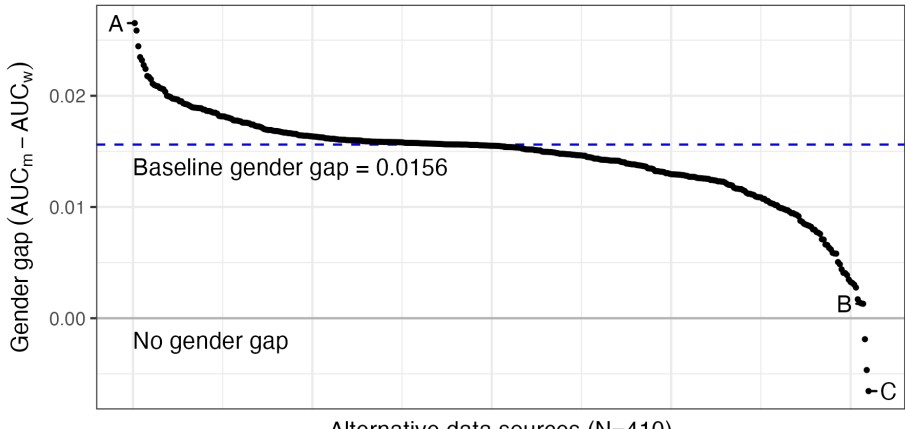

Figure 1: Distribution of gender gap from 410 alternative data sources. Each dot corresponds to an alternative data source used in the credit scoring algorithm. For each data source, the associated gender gap on the vertical axis is the average difference in the out-of-sample AUC based on 1,500 repeated subsamplings of the training data. The data source that results in the largest gender gap that disfavors women is the frozen nuggets and breaded meats (labeled as A). The data source that results in the smallest absolute gender gap is the chocolate confectionery category (labeled as B). The data source that results in the largest gender gap that favors women is the bar soap category (labeled as C).

which suggests that the data source contains a greater amount of creditworthiness signals for that gender.

While there may exist various approaches to quantify the amounts of creditworthiness signals within a given data source, we adopt the following operationalization. We compute the *absolute* value of the correlation coefficient between an individual's expenditure on each product subcategory within a given product category and their default status. This metric captures the strength of the correlations between the data source and creditworthiness, regardless of their directions. Given that a product category typically includes multiple subcategories, this approach can yield multiple correlation coefficients. For simplicity, we focus only on the coefficient with the largest absolute value for each data source. To calculate the gender difference in signals, we compute the correlation for men and women separately and take the difference for each product market.

Figure 2 illustrates the relationship between the gender disparity in the amounts of creditworthiness signals and the incremental gender gap across the 410 alternative data sources considered. The line of best fit in the graph indicates a positive association between data sources with uneven creditworthiness signals for a specific gender and a greater incremental gender gap. On the one hand, this result is reassuring, given that our definition of the gender gap is based on the gender difference in predictive accuracy. Therefore, leveraging a data source that contains more creditworthiness signals for men (women) is likely to lead to a positive (negative) incremental gender gap. On the other hand, we also find that while the algorithm is gender-blind (i.e., does not have gender as a direct input), many of the alternative data sources have differential impacts on men and women. This finding suggests

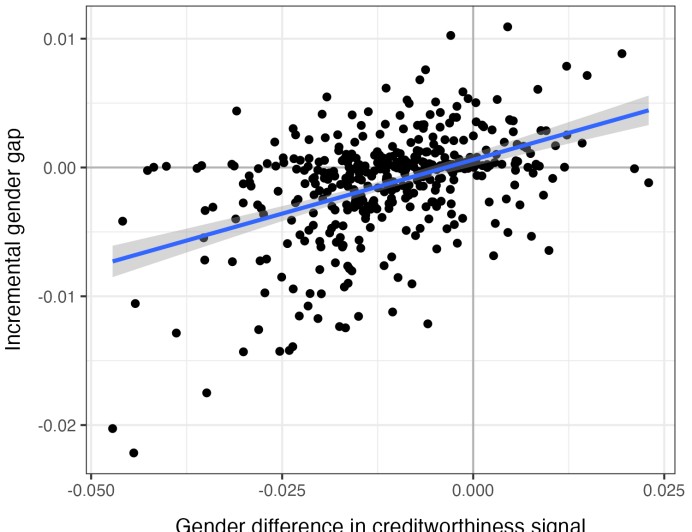

Figure 2: Relationship between signal disparity and incremental gender gap. Each dot corresponds to an alternative data source. The horizontal axis indicates the difference in the amounts of creditworthiness signals contained in a given data source for men and women. The blue line with a 95% confidence interval represents the line of best fit produced using Ordinary Least Squares (OLS) regression.

that the algorithm has the capability of recognizing the existence of distinct gender groups through proxy variables, which we examine shortly.

Also noteworthy is the fact that about 51% of the data sources (209 out of 410) are positioned in the third quadrant of the graph, which indicates that they contain a greater amount of creditworthiness signals for women compared to men. This empirical evidence underscores the potential of alternative data sources to benefit marginalized groups, such as women, by disproportionately improving predictive accuracy for them and therefore, closing the gender gap.

### 4.3.2 PROPERTY 2 (CLASS SEPARABILITY): CAPACITY TO PROXY GENDER

We now investigate the relationship between the class separability of data sources and the resulting incremental gender gap. Specifically, we train gender-predicting algorithms to quantify the proxy capability of data sources (Tschantz, 2022). These algorithms, similar to our credit scoring algorithms, predict gender using each of the 410 alternative data sources along with traditional ones. By comparing their predictive performance, measured by out-of-sample AUC, to algorithms trained solely on traditional data, we assess the incremental proxy capacity of alternative data sources.

Figure 3 describes the relationship between the incremental capacity to proxy gender and the incremental gender gap across the 410 alternative data sources. Panel (a) illustrates a positive relationship between the capacity to proxy gender and the *absolute* incremental gender gap. Simply put, a stronger proxy for gender has a larger absolute impact on the magnitude of the gap, regardless of whether it increases or decreases the gap. Further,

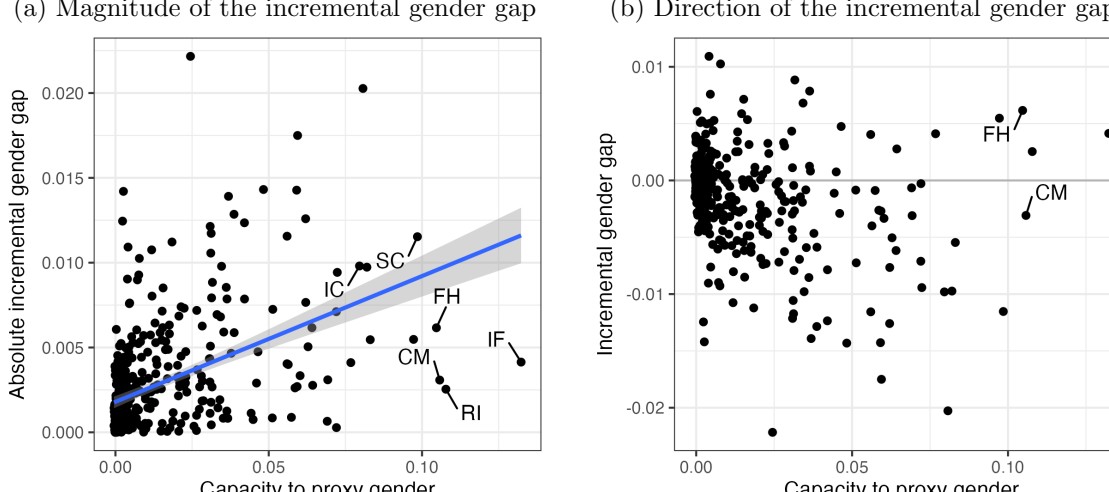

Figure 3: Relationship between capacity to proxy gender and incremental gender gap. In both panels, each dot corresponds to an alternative data source. The blue line represents the line of best fit produced using Ordinary Least Squares (OLS) regression, accompanied by a 95% confidence interval. The capacity to proxy gender on the horizontal axis is measured as the incremental out-of-sample AUC of gender-predicting algorithms that leverage both the corresponding alternative data source and the traditional data sources for the prediction. Panel A shows that stronger gender proxies tend to have a greater impact on the magnitude of the incremental gender gap (or absolute incremental gender gap), regardless of its direction. Panel B shows that gender proxies have a varying impact on the direction of the incremental gender gap.

the panel reveals interesting patterns regarding the nature of product categories. While spending on gendered products such as feminine hygiene items ("FH") strongly predicts gender, which is reassuring, expenditures on seemingly gender-neutral categories also show strong associations with gender. For example, spending on imported food ("IF"),[9] pantry staples like rice ("RI"), canned meat ("CM"), and instant coffee ("IC"), as well as necessities like shampoo and conditioner ("SC") exhibit strong correlations with gender.

One potential explanation for these findings is the gender disparity in purchasing patterns. Although we refrain from providing explanations for these behavioral differences, several factors may contribute to such variations. These include demand-side factors such as individual preferences, shopping objectives, and gender role norms, as well as supply-side factors like targeted promotions and the prevalence of gendered products within a category.

Panel (b) provides some insights into the *direction* of the incremental gap. Interestingly, we observe variations in the direction of the incremental gap among the data sources with similar proxying capacities for gender. For example, while data from the feminine hygiene products market ("FH") and data from the canned meats market ("CM") are similarly effective in predicting gender, the former leads to an increase in the gender gap, whereas

---

9. In the supermarket transaction data, there are five product categories dedicated to food items that are imported from different countries or regions. To ensure confidentiality, we have not disclosed their names.

the latter reduces it. This finding challenges the common assumption that data with a stronger proxying capacity for protected attributes would consistently disadvantage the minority group.

### 4.3.3 INTERACTION BETWEEN THE TWO PROPERTIES

To examine how the interplay between these two properties influences the direction and magnitude of the incremental gender gap, we regress the incremental gap associated with the 410 alternative data sources on their signal disparity measures (property 1) and class separability measures (property 2). Estimates are reported in Table 1. Columns (1) and (2) report the estimates when each data property enters the regression separately as an independent variable. The results confirm our findings in Figures 2 and 3(b): a positive association between the incremental gender gap and signal disparity (column (1)) and a slightly negative association between the incremental gender gap and class separability (column (2)).

In column (3), we find a positive and statistically significant coefficient on the interaction term between the two properties. In particular, the magnitude of the estimate (2.665) exceeds that of the signal disparity (0.105) by a factor of 25, while the estimate for the class separability is effectively zero in this specification. The standardized estimate for the interaction term (0.327) is also greater than that of the signal disparity term (0.285). These findings underscore that the interplay between these two properties plays a relatively large role in explaining the incremental gender gap. Specifically, among data sources with identical signal disparities, those with higher class separability contribute to a larger incremental gender gap. This suggests that even a slight imbalance in creditworthiness signals between genders within a data source can lead to a significant change in the gender gap if the data source effectively distinguishes between men and women. The reverse is also true: a data

| | DV: Incremental gender gap | | |
| | (1) | (2) | (3) |
| --- | --- | --- | --- |
| Signal disparity | 0.167 | | 0.105 |
| | (0.016) | | (0.020) |
| | | | |
| Class separability | | −0.063 | 0.005 |
| | | (0.009) | (0.015) |
| | | | |
| Signal disparity × Class separability | | | 2.665 |
| | | | (0.707) |
| | | | |
| Constant | 0.001 | −0.0004 | 0.0004 |
| | (0.0003) | (0.0002) | (0.0003) |

Table 1: Interaction between two data properties. Table reports the coefficients and standard errors in parentheses estimated from regressions where the dependent variable is the incremental gender gap, and independent variables are signal disparity (gender difference in predictive accuracy) and class separability (capacity to proxy gender) of an alternative data source. $N=410$.

source that is disproportionately predictive of creditworthiness for one group may have a minimal impact on the gap if its ability to proxy gender is weak.

## 4.4 Additional Analyses

### 4.4.1 Predictive Power and Gender Gap

We find that incorporating alternative data sources does not necessarily enhance the predictive power of credit scoring algorithms. In some cases, weak signals from these new data sets can introduce noise, reducing predictive accuracy even as the gender gap narrows. To illustrate this, we categorize the 410 alternative data sources along two dimensions: one represents the incremental gender gap, and the other captures the incremental predictive accuracy. We measure the incremental predictive accuracy for all consumers, as well as separately for men and women. The results are presented in Figure A.4 in the Appendix.

Two key observations emerge from this analysis. First, more than half of the 410 alternative data sources improve the predictive power of the algorithm (56% for the overall predictive accuracy, 58% for the gender-specific predictive accuracy for both men and women), while the rest result in lower predictive accuracy. This suggests a potential trade-off between different objectives of using alternative data sources for credit scoring: reducing the gender gap and improving predictive power.

Second, data sources that improve predictive power for the disadvantaged group, women, are most likely to reduce the gender gap, consistent with our earlier findings. The likelihood of reducing the gender gap varies significantly depending on whether predictive accuracy improves overall, for men, or for women. Specifically, we find that when predictive accuracy increases overall, there is a 52% chance that the gender gap decreases, and when it improves for men, this likelihood is 45%. However, when predictive accuracy improves for women, this likelihood rises significantly to 84%.

### 4.4.2 Behavioral Characteristics and Gender Gap

Beyond the two properties considered in this study, there can be several other ways to characterize alternative data. An alternative approach involves characterizing gender-specific purchase behaviors within each market, such as the number of consumers of each gender or the volume of data generated by each gender.

To assess whether and to what extent such behavioral characteristics can explain the incremental gender gap, we create a series of gender-specific behavioral variables for the 410 data sources and examine their correlations with the incremental gender gap. Our regression analysis (reported in Table A.4 in the Appendix) reveals an interesting contrast. The number of female shoppers in a given market, which simply captures the presence of female shoppers, is negatively and statistically significantly associated with the incremental gender gap. In other words, data from markets with a larger number of female shoppers are more likely to reduce the gender gap. In contrast, variables that capture the gender-specific intensity of shopping, such as shopping frequency and spending amounts, do not show significant associations with changes in the gender gap.

## 5 Discussion

Fair lending laws, such as the Equal Credit Opportunity Act and the Fair Housing Act in the United States, not only prohibit the explicit consideration of protected attributes during credit evaluations (referred to as disparate treatment) but also deem it unlawful to implement policies or practices that disproportionately burden individuals in protected classes (referred to as disparate impact). Even unintentional proxy discrimination can fall within the scope of disparate impact.

In the evolving landscape of credit markets, however, lenders are increasingly gaining access to extensive consumer data. This shift is driven by the emergence of data markets that facilitate data transfers across companies and across domains, as well as the growing participation of big tech companies equipped with first-party consumer data in the financial services sector (e.g., Amazon Pay and Apple Card). These trends raise concerns about the possibility that lenders might leverage a combination of features derived from large-scale and high-dimensional data to construct strong proxies of protected attributes. Indeed, a conventional belief is that the use of strong proxies for protected attributes in decision-making would generally exacerbate existing disparities (e.g., Barocas and Selbst, 2016; Prince and Schwarcz, 2019; Tschantz, 2022).

We investigate the association between data properties and the resulting disparate impact, an area that remains largely unexplored. By studying this nuanced interaction, we attempt to shed light on how the use of proxies can yield contrasting effects, either exacerbating or mitigating inequalities, and identify conditions under which strong proxies of protected attributes can advantage the minority group, women. Specifically, our findings suggest that incorporating strong gender proxies into the credit scoring process has the potential to reduce the gender gap in credit risk prediction accuracy and, potentially, in credit allocation, provided that the data contain disproportionately larger amounts of creditworthiness signals associated with women. We believe this finding has significant relevance to the question of how the utilization of alternative data should be regulated in consumer credit markets.

One might question the feasibility of using retail data or any non-traditional data as alternative data due to privacy concerns. One potential solution involves allowing consumers to voluntarily submit their data. In several countries, such as the United States, the European Union, and the United Kingdom, individuals can opt to share their detailed transaction and payment data with third-party financial service providers through "open banking" (He et al., 2023). Another consideration is the ethical and fairness implications of denying credit based on alternative data. In response, the US has adopted the "second look" approach, which leverages alternative data only to approve applicants who would have been denied using traditional data, rather than rejecting those who would have been approved under traditional criteria.

Our findings may have broader implications, as concerns about proxy discrimination and algorithmic fairness extend beyond credit markets. For example, consider the recent U.S. Supreme Court ruling on affirmative action in higher education. While the ruling prohibits the use of race variables in college admission, there is concern that facially race-neutral variables, such as standardized test scores, may serve as strong proxies for race (e.g. Neal and Johnson, 1996; Card and Rothstein, 2007; Goodman et al., 2020). Given that this study

focuses on data from a specific domain and its impact on gender inequalities, it remains uncertain whether and to what extent certain racial proxies might mitigate or exacerbate inequalities in college admission across racial groups. Future studies exploring such contexts would be valuable in enriching our understanding of the impact of proxy variables.

### 5.1 Limitations and Future Work

We acknowledge several limitations of this study that provide avenues for future research. First, our analyses rely on simulating credit scoring processes using data solely from approved applicants. We lack access to information on rejected applicants as well as individuals who have not applied, possibly because they are not targeted by lenders or are outside the formal financial system. This absence of data on excluded individuals precludes us from assessing the external validity of our findings for these consumers. To better understand the impact of leveraging alternative data across the broader population, it would be essential to obtain data on these excluded individuals, ideally through an experiment that randomizes credit extensions.

Further, we recognize that beyond the two data properties considered in our study, there may be a wide range of other properties that could affect the gender gap. Although this paper does not attempt to identify all potential determinants of the gender gap or quantify their relative impacts, a comprehensive understanding of these properties would be invaluable. Another important determinant of gender disparities in credit access is *how* credit scores or predictions are integrated into the decision-making process. Different lenders, even when presented with the same set of predictions, may apply varying decision rules, leading to different credit decisions and distribution. Although our paper remains agnostic about the specific decision-making framework of lenders, it would be beneficial if future studies could investigate the role of the decision-making process in disparities in credit access.

Lastly, this study focuses on a particular form of alternative data–individuals' supermarket expenditure–for the purpose of credit scoring. One obvious advantage of this data lies in its inclusiveness: grocery shopping at a supermarket is a universal activity that transcends socioeconomic and cultural boundaries. It is reassuring to find that, even within our seemingly homogeneous supermarket data, there is considerable heterogeneity across different product categories in their impact on the gender gap. This variation supports our argument about the importance of the two key properties of data sources in influencing the gender gap. However, we recognize that supermarket transaction data may not fully capture the range of alternative data sources. Exploring other sources, such as durable goods purchase data or phone usage data, could reveal different impacts on the gender gap. We note that future research could expand its scope to consider data from other domains and document their impact on algorithmic fairness, not only in credit scoring but also in other types of predictions and decisions in the consumer finance sector.

## 6 Conclusion

In this paper, we leverage a unique data set to explore the potential disparate impact of leveraging alternative data in the consumer finance context. We identify two data properties, signal disparity and class separability, and empirically demonstrate that the properties can

at least partially explain the resulting gender gap in the predictive accuracy of credit risks. Our findings are particularly relevant to settings where certain user attributes are protected, and therefore, not accessible by firms or researchers. We hope that this paper serves as a case study that explores an important societal issue around the increasing use of new data and advanced prediction algorithms.

## Broader Impact Statement

Our findings may have significant policy implications for the regulation of alternative data in credit markets. For example, it raises questions about whether a blanket prohibition on the use of gender proxies or proxies for any protected group membership is appropriate for promoting financial inclusion without exacerbating existing inequalities. We acknowledge that the implementation of this idea may have contrasting effects on the distribution of credit, depending on how lenders integrate the data sources into their decision-making processes. For example, using alternative data sources to approve applicants who would have been otherwise rejected under the traditional lending criteria, rather than rejecting applicants who would have been otherwise approved, can help mitigate potential fairness concerns and promote greater credit access among a wider spectrum of individuals. Beyond lending decisions, the core idea of this paper has broader applications in areas such as college admission decisions, hiring decisions, and criminal justice decisions. These are all domains where increasingly extensive data is utilized to assess an individual's qualifications, posing risks of proxy discrimination and algorithmic bias. Future studies exploring these contexts would be valuable in enriching our understanding of the impact of proxy variables.

## Acknowledgments and Disclosure of Funding

The authors are grateful to the Wharton Customer Analytics and an anonymous data provider for sharing data and domain knowledge. The initial draft of this work was reviewed by the anonymous data provider in accordance with the data use agreement. The data provider did not request any revisions or changes. The authors declare that they have no financial interests that relate to the research described in this paper. This work was previously circulated under the title "Leveraging Gender Proxies Can Lead to Fairer Credit Risk Predictions." The authors thank Eric Anderson, Ayelet Israeli, Garrett Johnson, Sora Jun, Tai Lam, and Anna Tuchman for helpful discussions and feedback.

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

## Appendix A. Additional Tables and Figures

Table A.1: Product hierarchy in the supermarket data

| Section | Product Category (number of unique categories) |
|---|---|
| Accessories | Bag and purse; Belt and suspender; Fashion jewelry; Hat and cap; Other accessories; Scarf and shawl; Tie and bow tie (7) |
| Baby | Baby accessories; Baby outerwear (0-36 months); Big nursery; Nightwear (0-36 months); Sets (0-36 months); Small nursery; Socks and shoes (0-36 months); Sportswear and swimwear (0-36 months); Top (0-36 months); Trousers and dress (0-36 months); Underwear (0-36 months) (11) |
| Bakery and Pastry | Fresh white bread; Fresh special bread; Local bread; Viennese products (4) |
| Beverage | Carbonated drinks; Energy drinks; Healthy drinks; Instant powder drinks; Mineral water; Non-carbonated drinks; Sparkling water; Syrups; Vitamin sport drinks; Water (10) |
| Butchery | Beef; Butchery service counter; Lamb; Mutton; Offal; Poultry; Prepared meats; Veal (8) |
| Camping Gardening | Camping; Garden furniture; Gardening (3) |
| Car | Car accessories: interior; Car accessories: exterior; Car cleaning; Maintenance; Motorcycle (5) |
| Children | Boys' bottom (2-14 years); Boys' nightwear (2-14 years); Boys' outerwear (2-14 years); Boys' sportswear (2-14 years); Boys' underwear (2-16 years); Boys' top (2-14 years); Dress and skirt (2-14 years); Girls' bottom (2-14 years); Girls' nightwear (2-14 years); Girls' outerwear (2-14 years); Girls' sportswear (2-14 years); Girls' top (2-14 years); Girls' underwear (2-16 years); Set (2-14 years); Socks (2-16 years) (15) |
| Cigarettes | Cigarettes (1) |
| Coffee Shop | Drinks; Hot drinks; Pastry desserts; Sandwiches menus (4) |
| Dairy Counter | Dairy product counter; Local hard cheese; Organic; Speciality; Western cheese (5) |
| Dairy Products | Butter; Cheese of the world; Cheese spread and processed cheese; Dairy UHT milk; Eggs; Fresh whey cheese; Margarine and spreads; Natural sliced cheese; Organic dairy products; Powdered and shredded cheese; Specialty UHT milk; White cheese (12) |

Table A.1: Product hierarchy in the supermarket data

| Section | Product Category (number of unique categories) |
|---|---|
| Delicatessen | Meat slices; Mortadella; Pasta; Pastry; Potted meat; Ready-to-eat; Sausages; Seafood; Smoked, salted, and dry fish; Vegan (10) |
| Delicatessen Counter | Catering: homemade; Catering: purchased; Convenience counter; Cooked and roasted products; Delicatessen product; Fish products; Food-to-go; Meat products; Semi-consignment (9) |
| Do-It-Yourself | Batteries; Electrical bulbs; Hardware accessories; Hardware tools; Home electronics; Insulation hardware; Power tools; Power washers; Sockets, cables, and plugs; Storage; Torch, lamp, and emergency light (11) |
| Fishery | Cooked fish; Fresh water fish; Lobster and shrimp; Pre-packed seafood; Saltwater fish; Seafood delicatessen; Shellfish and mollusks; Smoked and dried fish (8) |
| Frozen Food | Bulk ice cream; Burgers; Ice cubes and bags; Fish and seafood; Franks; Fried vegetables; Frozen cooking fats; Frozen desserts; Frozen pastries; Frozen nuggets and breaded meats; Minced meats and meatballs; Meat parts; Pizza; Ready and prepared meals; Single-portion ice cream; Soups and starters; Vegetable (17) |
| Fruits and Vegetables | Dried vegetables in bulk; Dry commodities; Flowers plants; Fresh fruits; Fresh juices; Fresh vegetable; Organic; Ready-to-eat: IV gamme; Ready-to-eat: V gamme (9) |
| Gift Shop | Clocks; Watches (2) |
| Grocery | Baby food; Baking ingredients; Biscuits; Candy; Canned fish; Canned fruits and raw nuts; Canned meats; Canned vegetables; Cereals; Chips, crisps, and puffs; Chocolate confectionery; Cooking ingredients; Cooking oil and ghee; Dry breads; Diet and fitness; Imported food: A; Imported food: B; Imported food: C; Imported food: D; Imported food: E; Evaporated creamers; Flours; "Free" products (e.g., GMO-free); Gumdrops; Honey; Instant coffee; Jam; Noodles; Nuts and seeds; Olive oil; Olives and pickles; Organic and bio: salty; Organic and bio: sweet; Pasta; Pet food; Powder drinks; Powder milk; Pulses; Rice; Roasted coffee; Salt, pepper, and spice; Sauces; Seasonal; Soups; Spreadable; Sugar; Tea; Vinegar dressing (48) |
| Hi-fi Sound | Audio portable devices; Car radio; Hi-fi system; Home theater; Radio K7 recorder; Separated components; Small audio; Sound accessories (8) |
| Home Linen | Bathrobe; Bathroom linen; Bed linen; Blanket; Children; Comforter; Decoration; Haberdashery; Housemaid uniform; Kitchen linen; Mat; Mattress cover; Pillow (13) |
| House Equipment | Carpets, doormats, and rugs; Furniture storage; House decoration; Lighting; Sanitary (5) |
| Household Appliances | Air conditioning; Built-in; Cookers; Dishwasher; Freezer; Fridge; Washers and dryers (7) |
| Household Goods | Breakfast; Clothes care; Cooking; Floor maintenance; Food preparation; Hair care; Hygiene beauty; Microwave and oven; Shavers; Spare parts; Well-being (11) |
| Houseware | Casserole; Cleaning cloth; Cooking; Disposable items; Frying pan; Glassware; Kitchenware; Food saver and storage; Shisha; Tableware (10) |
| Ladies | Knitwear and sweater; Ladies' traditional wear; Nightwear; Outerwear; Shirt and blouse; Skirt and dress; Socks; Sportswear; Swimwear; Trousers; T-shirt and polo; Underwear (12) |
| Library | Bestsellers; Children's books; English books; Local books; Newspapers and magazines (5) |
| Luggage | Bag; Business bag; Children's trolley; Outdoor bag; Suitcase; Trolley (6) |
| Men | Knitwear and sweater; Men's socks; Men's traditional wear; Men's underwear; Nightwear; Outerwear; Shirt; Sportswear; Trousers; T-shirt and polo (10) |
| Mobility | Calculators; Fixed line; GPS; Phone cards; Phones; Phone accessories (6) |

Table A.1: Product hierarchy in the supermarket data

| Section | Product Category (number of unique categories) |
|---|---|
| Non-food Grocery | Air fresheners; Baby toiletries; Baby wipes and detergent; Bar soap; Bathroom toilet papers; Beauty: organic; Beauty: special; Birth control and family planning; Cotton pads; Detergent: liquid tabs; Detergent: powder; Diapers; Dish washing; Fabric care; Fabric softener; Feminine hygiene products; Foot care; Green: home; Green: baby; Garbage bags; Hair accessories; Hair coloration; Hair fixation; Hair treatment; Home cleaning: specific; Home cleaning: multi-purpose; Kids' beauty; Kitchen disposables; Kitchen towels; Liquid soap; Makeup accessories; Men's deodorant; Men's grooming; Men's perfume; Oral care; Parapharmaceutical products; Shampoo and conditioner; Shower gel; Skincare; Sponges; Sun care products; Tissues and wipes; Women's deodorant; Women's perfume; Women's depilatory (45) |
| Office Automation | Digital cards; Gaming; Media recording; Microcomputer; PC accessories; PC peripherals; PC software; Printing (8) |
| Pastry | Cakes; Dry; Individual pastry; Special pastry; Traditional counter (5) |
| Photo | Binoculars; Camcorder; Camera accessories; Cameras; Memory card; Printer (6) |
| Poultry | Chicken; Cooked and uncooked poultry; Giblets; Others; Turkey (5) |
| Shoes | Baby shoes; Beach shoes; Boys' shoes; Children's sport shoes; Girls' shoes; Lace; Men's shoes; Men's sport shoes; Women's shoes; Women's sport shoes (10) |
| Sports | Bicycle; Fishing; Fitness; Individual sports; Sports Bag; Team sports; Water sports (7) |
| Stationery | Drawing stationery; Greeting cards; Lunch drinks; Notebooks; Office filing stationery; School bags; School material; Writing equipment; Writing stationery (9) |
| Toys | Babies' toys; Boys' toys; Girls' toys; Seasons greetings decoration; Summer games; Unisex toys (6) |
| TV and VCR | Laser disc and video; Projector; Satellite antenna; TV; TV accessories (5) |
| Ultra Fresh | Chilled cakes; Chilled desserts; Fresh cream; Fresh juice; Fresh milk; Industrial bread; Kids; Labneh; Non-chilled pastries; Organic health; Sour milk products; Yogurts (12) |

| | Min | Q1 | Median | Mean | Q3 | Max |
|---|---|---|---|---|---|---|
| Number of product subcategories | 1.00 | 3.00 | 5.00 | 5.33 | 7.00 | 19.0 |
| Number of brands in the category | 1.00 | 4.00 | 11.0 | 19.9 | 27.0 | 221 |
| Average category purchase frequency | 0.00 | 0.06 | 0.29 | 1.06 | 1.01 | 37.0 |
| Average category spending amount (USD) | 0.00 | 1.72 | 5.56 | 15.6 | 1,574 | 660 |

Table A.2: Summary statistics of product categories (alternative data sources). Unit of observation is a product category, and values are based on the 410 product categories in the supermarket data. The average purchase frequency and the average spending amount are calculated based on the purchases made by the final sample of 30,005 individuals during the sample period (Period 1 of our empirical design in Section 3.2).

| Variables | Type | Number | Example values |
|---|---|---|---|
| Number of dependents | Categorical | 7 | 0, 1, ..., 5, 6+ |
| Employment status | Categorical | 3 | Salaried, Self-employed, Other |
| Occupation | Categorical | 4 | Manager, Officer, Other, Invalid |
| Monthly income (USD) | Continuous | 1 | $4,710, $5,299, ... |
| Credit score | Categorical | 11 | Unknown, Decile 1, ..., Decile 10 |

Table A.3: Features used in a baseline credit scoring algorithm. Table reports the features derived from traditional data sources, which are used to build a baseline credit scoring algorithm. In order to include individuals with missing credit scores into the algorithm training process, we discretize the credit score variable. Specifically, for individuals without credit scores, we create a category called "Unknown" and place their credit score values (NA) with the category. For those with credit scores, we assign them into one of the ten income-decile bins.

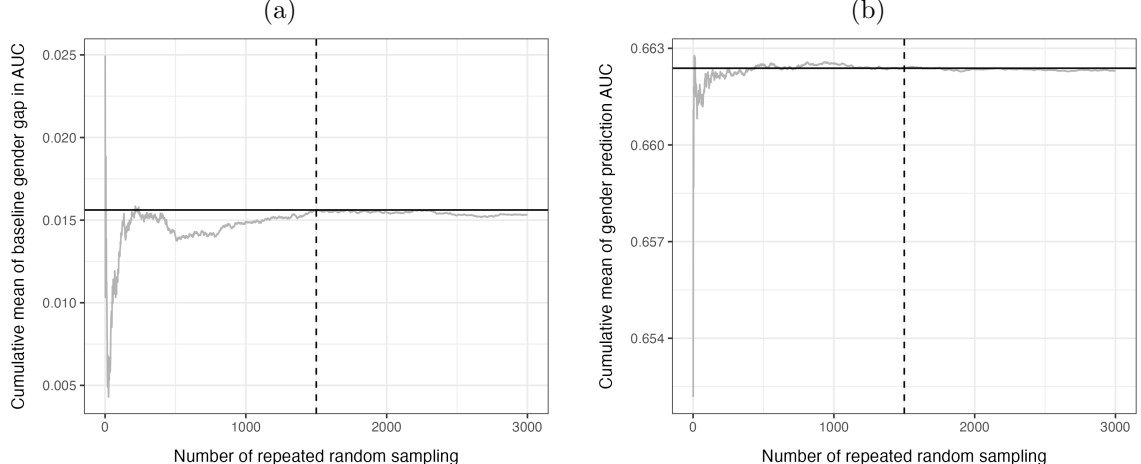

Figure A.1: Repeated random sampling of training data. Panel A reports the cumulative mean of the baseline gender gap in terms of the out-of-sample AUC as the number of random samples increases. Panel B reports the cumulative mean of the out-of-sample AUC of the gender-predicting algorithms as the number of random samples increases. The solid horizontal line indicates the mean value of the first 1,500 random samples, and the dashed vertical line represents the position of the 1,500th sample. The cumulative mean becomes stabilized at around the 1,500th sample.

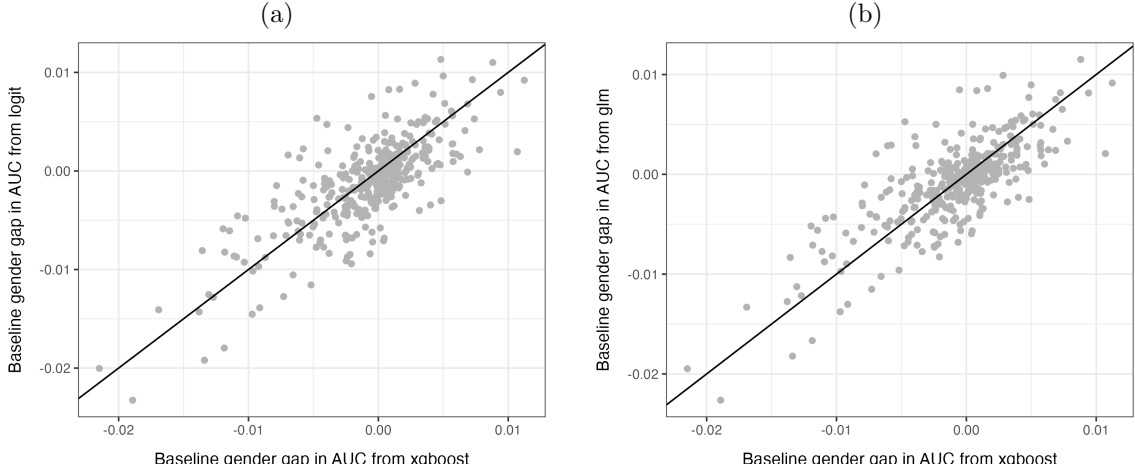

Figure A.2: Comparison to alternative classifiers. In both panels, each dot represents an alternative data source. Panels compare the incremental gender gap that results from the use of XGBoost as a classifier (horizontal axis) with the incremental gender gap that results from the use of two alternative binary classifiers (vertical axis). Panel A compares the incremental gender gap against the logistic regression model, and Panel B against the elastic-net regularized generalized linear model. The incremental gender gap introduced by the two alternative classifiers is averaged across 500 repeated random splits of training and test sets. For easier interpretation, we provide a 45-degree line as a point of reference: the closer the dots are to the line, the smaller the difference in the resulting gap between XGBoost and the alternative classifier.

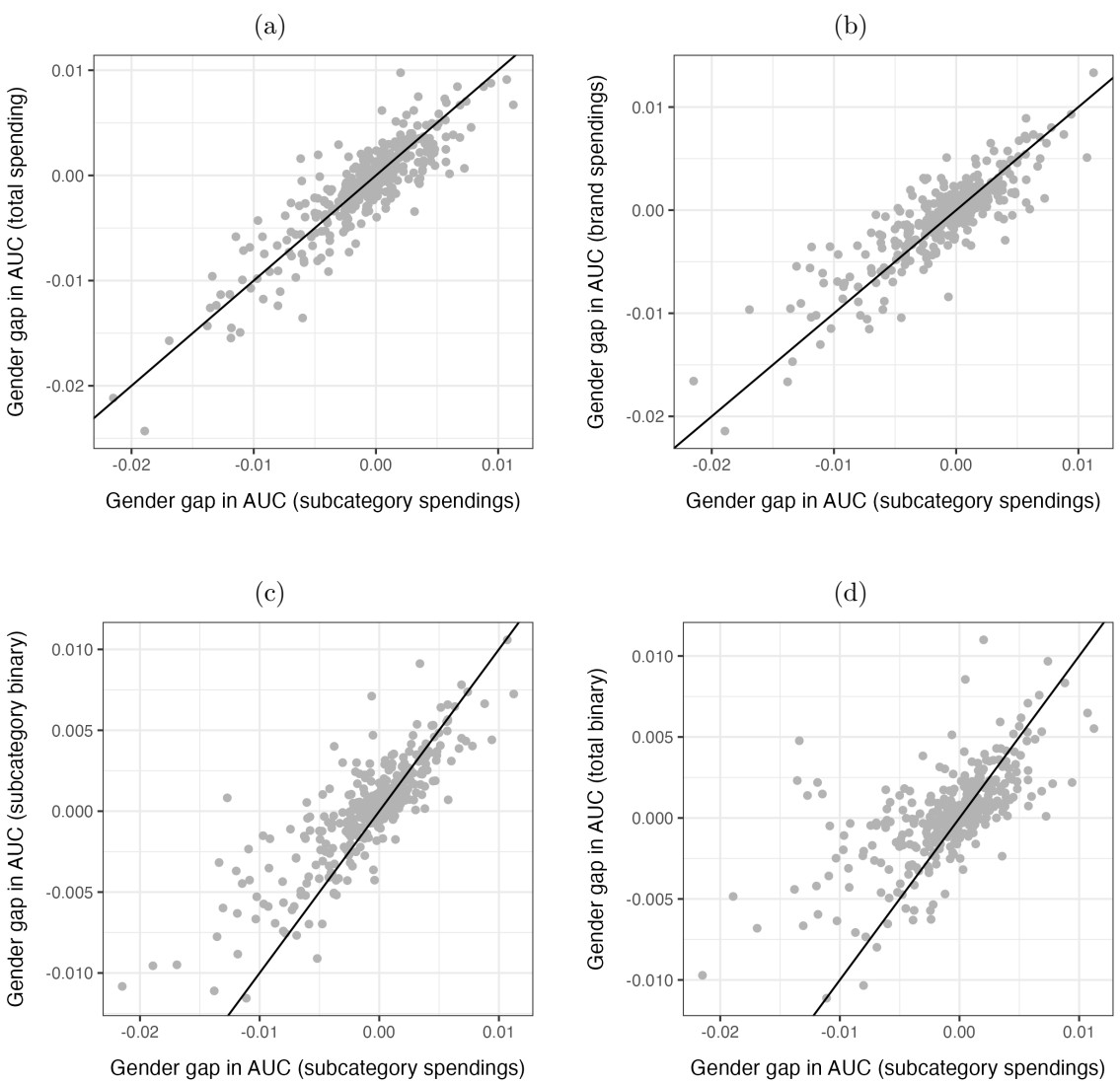

Figure A.3: **Comparison to alternative feature engineering approaches.** In all panels, each dot represents an alternative data source. Panels compare the incremental gender gap based on our primary approach of feature engineering the supermarket data (horizontal axis) with the incremental gender gap based on four alternative approaches of feature engineering (vertical axis). Panel A: use an individual's total expenditure on the corresponding product category, Panel B: use brand-level expenditure within the corresponding product category, Panel C: use a set of binary indicators of whether an individual purchased each of the subcategories within the corresponding product category, Panel D: use an indicator of whether an individual purchased the corresponding product category. The incremental gender gap introduced by the four alternative approaches is averaged across 500 repeated random splits of training and test sets. For easier interpretation, we provide a 45-degree line as a point of reference: the closer the dots are to the line, the smaller the difference in the resulting gap between our primary and the alternative feature engineering approaches. The figure demonstrates a strong correlation in the resulting incremental gender gap between our primary approach and the four alternative approaches considered, which indicates that our findings are robust to the feature engineering approach.

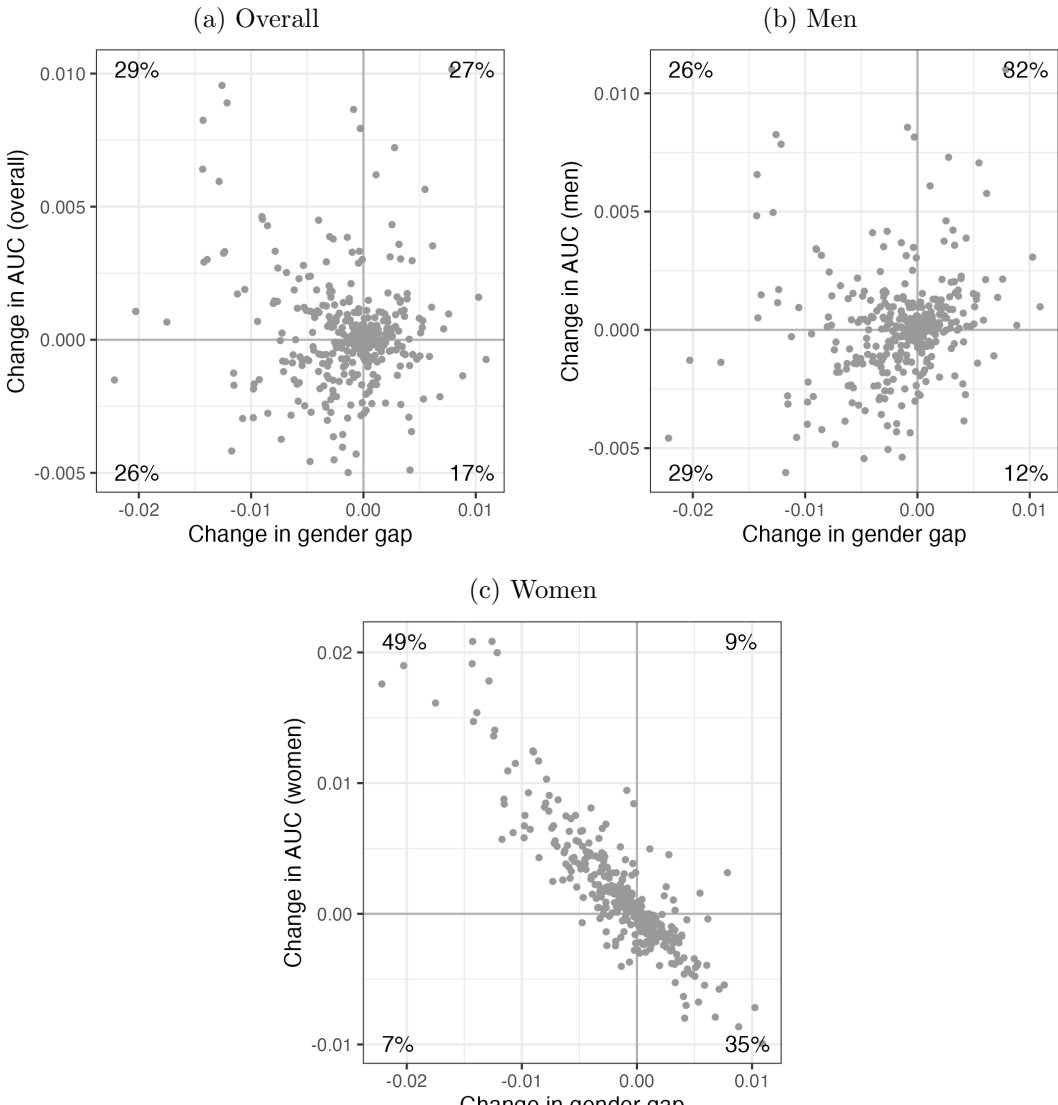

Figure A.4: Changes in predictive power and gender gap. In all panels, each dot corresponds to an alternative data source. The change in gender gap is on the horizontal axis, whereas the change in predictive power is on the vertical axis. The predictive power is measured as the incremental out-of-sample AUC of gender-predicting algorithms that leverage both the corresponding alternative data source and the traditional data sources for the prediction.

|  | DV: Incremental gender gap | | | |
|  | (1) | (2) | (3) | (4) |
| --- | --- | --- | --- | --- |
| No. male shoppers (in 1,000s) | 0.0001 | −0.00003 | | |
|  | (0.0001) | (0.0001) | | |
| No. female shoppers (in 1,000s) | −0.002*** | −0.002*** | | |
|  | (0.001) | (0.001) | | |
| Share of female shoppers | | | −0.005 | −0.002 |
|  | | | (0.004) | (0.004) |
| Avg. no. items purchased by male (in 1,000s) | | 0.446 | | 0.149 |
|  | | (0.559) | | (0.585) |
| Avg. no. items purchased by female (in 1,000s) | | −0.288 | | 0.284 |
|  | | (0.517) | | (0.542) |
| Avg. trip frequency by male (in 1,000s) | | −0.331 | | −0.118 |
|  | | (0.799) | | (0.860) |
| Avg. trip frequency by female (in 1,000s) | | 0.466 | | −1.210 |
|  | | (0.776) | | (0.806) |
| Avg. spending amount by male (in 1,000s) | | 0.001 | | 0.001 |
|  | | (0.001) | | (0.001) |
| Avg. spending amount by female (in 1,000s) | | −0.001 | | −0.0005 |
|  | | (0.001) | | (0.001) |
| Constant | 0.001** | 0.0003 | −0.0004 | 0.001 |
|  | (0.0003) | (0.0004) | (0.001) | (0.001) |

Table A.4: Behavioral characteristics and gender gap. Table reports the coefficients and standard errors (in parentheses) estimated from regressions where the dependent variable is the incremental gender gap, and independent variables are behavioral characteristics of male and female shoppers within each product market. $N$=410.

