# OpenReview forum: "Properties of Alternative Data for Fairer Credit Risk Predictions"
_DMLR — Accepted by DMLR_

### Review · Reviewer_YCdP · 2024-07-11

**Recommendation:** 4
**Confidence:** 3

**Summary Of Contributions:**

This study examines the effect of alternative data on gender disparities in credit scoring. The authors use supermarket transaction data, categorized into 410 distinct product groups, as a novel source of alternative information. To simulate credit scoring, they develop binary classifiers to predict whether the customers will default on their payments and analyze how gender gaps change when incorporating this alternative data. The research identifies two key factors influencing gender gap changes: the varying amounts of creditworthiness signals in different data sources and their capacity to proxy gender. The authors support their hypotheses with empirical evidence.

**Strengths:**

Please see the strengths listed as above.

**Audience:**

Yes

**Broader Impact Concerns:**

The authors has included a broader impact statement at the end of the main paper.

**Claims And Evidence:**

The claims in the submission are well-supported by accurate, convincing and clear evidence.

**Datasets And Benchmarks:**

N/A

**Extended Submissions:**

N/A

**Limitations:**

Please see the above weaknesses.

**Requested Changes:**

1. The transaction data are retrieved from the retailer's loyalty membership program. However, is it possible that all the household members may share the same loyalty membership card? For example, the wife may use her husband's membership card for shopping. In that case, the gender information linked to the product transaction data will be inaccurate. The authors should explain how this gender noise may affect the evaluation result.

2. I recommend the authors to include a separate section to state how one can access their data and reproduce the simulations. I would greatly appreciate it if the authors would release their data and code.

**Strengths And Weaknesses:**

**Strengths**
This work makes several notable contributions to interpret how the inclusion of alternative data may influence the gender gap in credit scoring.

Firstly, I would appreciate the authors if they can release the data they used for the simulation. It is a well-known issue that there lacks reliable data sources for credit scoring in the algorithmic fairness community. The inclusion of retail purchase records as alternative data is novel.

Secondly, this paper is well-organized and clearly articulate how they run the simulations. The evaluation metric, the gender gap of AUCs, is decent. The characterization of signal disparity and class separability is reasonable, although I would like to see more analysis from a theoretical perspective.

Finally, this work has identified the relation to the prior work. I like their justification on whether  the transaction and payment data is accessible in the real world. I also appreciate the authors has noted the limitations of their simulation.

**Weaknesses**
The alternative data is retrieved from the purchases made through the retailers' loyalty membership program. In addition, the data set is extremely imbalanced for different groups, with 86% men and only 14% women. The authors failed to address the potential selection bias in the data collection process, which may further exaggerate the gender gap.

---

### Review · Reviewer_4QxR · 2024-07-17

**Recommendation:** 3
**Confidence:** 1

**Summary Of Contributions:**

This paper proposes to use alternative data that are traditionally not used by financial institutions for credit scoring process to close the gender gap where men are favored by women despite evidence suggests that women are better at repaying. The alternative data (this paper uses supermarket purchase records) can have more accurate creditworthiness signals, which will lead to fairer ML model predictions after we incorporate such signals. The authors further demonstrate a positive correlation between gender-differential creditworthiness signals from the data source (410 products) and the predictive gender gap, and a positive correlation between the data source (410 products)’s capacity to proxy gender and the magnitude (not the direction) of the predictive gender gap. The authors identify that a small difference on creditworthiness signals between genders within a data source can lead to a significant change in the gender gap if the data source can distinguish between men and women (strong capacity to proxy gender).

**Strengths:**

All strengths of this paper are listed above in the Strengths and Weaknesses section.

**Audience:**

Yes

**Broader Impact Concerns:**

There is a Broader Impact Concerns section in the paper. The discussion of legislative implication by this paper (e.g. appropriateness of unconditional prohibition on the use of gender proxies, fairness or privacy converns on adopting alternative data) is elaborated. The applications beyond credit scoring domain are also provided (e.g. college admission, hiring). The ethical implications of this paper are overall well-discussed.

**Claims And Evidence:**

The main focus of this paper is to use alternative data to alleviate the predictive gender gap. The authors first show there is a positive correlation between gender-differential creditworthiness signals and incremental gender gap (Figure 2), and a positive correlation between the magnitude / absolute value of the incremental gender gap and the capacity to proxy gender (Figure 3a). The statistical significance calculation (Table 1) further demonstrates the joint / amplification effect between signal disparity and class separability. This evidence chain should be sufficient to support the central claim.

**Datasets And Benchmarks:**

High-level description and the data source of the dataset are disclosed. This dataset is not publicly available and will have difficulty for a full-scale experiment reproduction. The authors only  provide a summary of dataset that includes the product category hierarchy and the summary statistics of product categories.

**Extended Submissions:**

This paper is not an extended submission. To the best of my knowledge, there is no similar publication to this paper.

**Limitations:**

There are no other limitations. All weaknesses are listed above in the Strengths and Weaknesses section.

**Requested Changes:**

1. The results and analyses of absolute (gender-specific) prediction performance after incorporating each product category should be added.

2. The discussion of the cause of disproportionate creditworthiness signals that are present in the alternative data but not in the data that traditional credit scoring process would use should be provided.

**Strengths And Weaknesses:**

Strengths:
1. The high-level idea is clear and well-motivated. The writing is overall clear and paper is well-structured.

2. The cause of incorporating alternative data would have the potential to alleviate the gender gap is well discussed (from the persepctive of creditworthiness signals and class separability in Sec 4.3).

Weakness:
1. Does not justify that decreasing the original predictive gender gap is not at the cost of decreasing the original prediction performance. The results of *absolute* prediction performance after incorporating alternative data from each of 410 product categories are not provided, and such analysis is also not present in the paper. The absolute (gender-specific) prediction performance is also crucial for the paper analysis.

2. Does not provide sufficient analysis for the cause of disproportionate creditworthiness signals that are present in the alternative data (supermarket transaction records, etc.) but not in the data that traditional credit scoring process would use (repayment history, etc.). I believe such analysis can provide insights on which data source would have the potential to narrow the predictive gender gap.

Disclaimer: I am not a domain expert in credit scoring and I am unfamiliar with the usual practices of credit scoring and the use of alternative data.

---

### Review · Reviewer_F9Fh · 2024-07-24

**Recommendation:** 3
**Confidence:** 2

**Summary Of Contributions:**

The paper explores the potential of using alternative data source for the gender gap problem in credit risk prediction. It leverages different sources of alternative data, analyzes how they impact the gender gap, and summarizes two properties of these data sources that affect the outcome.

**Strengths:**

- Using alternative data for gender gap seems to be an important and practical approach to the problem.
- Instead of using only a single source of alternative data, the paper uses 410 different data sources and analyzes how they affect the gender gap in more fine-grained way.
- The paper analyzes why certain alternative data sources help closing the gender gap while others widening it, and summarizes two properties. They also analyze the interactions between the two properties, which is insightful.
- The presentation is clear and easy to understand.

**Audience:**

Yes

**Broader Impact Concerns:**

The paper has included them. No more concerns from my side.

**Claims And Evidence:**

Yes.

**Datasets And Benchmarks:**

Not apply.

**Extended Submissions:**

No

**Limitations:**

- Although there are 410 product categories as different alternative data sources, they are still all supermarket data which lacks diversity to some extent.
- The two properties are summarized manually and it is unclear whether they are comprehensive or not. It would be better if we have some automatic ways to discover such properties.

**Requested Changes:**

None

**Strengths And Weaknesses:**

See the Strengths and Limitation parts.